# Mometasone Furoate in Non-Allergic Rhinitis: A Real-Life Italian Study

**DOI:** 10.3390/jpm12071179

**Published:** 2022-07-20

**Authors:** Angela Rizzi, Giuseppe Parrinello, Eugenio De Corso, Laura Tricarico, Michele Centrone, Alessia Di Rienzo, Chiara Laface, Giulio Cesare Passali, Gabriella Cadoni, Riccardo Inchingolo, Gaetano Paludetti, Jacopo Galli, Eleonora Nucera

**Affiliations:** 1UOSD Allergologia e Immunologia Clinica, Dipartimento Scienze Mediche e Chirurgiche, Fondazione Policlinico Universitario A. Gemelli IRCCS, 00168 Rome, Italy; angela.rizzi@policlinicogemelli.it (A.R.); p.giuseppe@email.it (G.P.); michelecent87@hotmail.it (M.C.); alessiadirienzo@libero.it (A.D.R.); chiara.laface01@gmail.com (C.L.); eleonora.nucera@policlinicogemelli.it (E.N.); 2Otorhinolaryngology Unit, Head and Neck Department, Fondazione Policlinico Universitario A. Gemelli IRCCS, 00168 Rome, Italy; eugenio.decorso@policlinicogemelli.it (E.D.C.); lauratricarico90bis@gmail.com (L.T.); giuliocesare.passali@unicatt.it (G.C.P.); gabriella.cadoni@unicatt.it (G.C.); gaetano.paludetti@unicatt.it (G.P.); jacopo.galli@policlinicogemelli.it (J.G.); 3Otolaryngology Institute, Università Cattolica del Sacro Cuore, 00168 Rome, Italy; 4UOC Pneumologia, Dipartimento Scienze Mediche e Chirurgiche, Fondazione Policlinico Universitario A. Gemelli IRCCS, 00168 Rome, Italy; 5Medicina e Chirurgia Traslazionale, Università Cattolica del Sacro Cuore, 00168 Rome, Italy

**Keywords:** non-allergic rhinitis, intranasal corticosteroid, mometasone furoate, quality of life, nasal cytology, olfactory function, fiberoptic nasal endoscopy, non-allergic rhinitis with eosinophils (NARES), the rhinitis quality of life questionnaire (RQLQ)

## Abstract

Background: In order to evaluate the efficacy of intranasal mometasone furoate in patients with non-allergic rhinitis (NAR), a real-life, observational, prospective study is performed. Methods: Thirty-one patients (age 18–64 years) receive intranasal (mometasone furoate, 200 µg b.i.d. for 15 consecutive days per month for 6 consecutive months), plus isotonic nasal saline. The cytologic pattern of local inflammation, nasal airflow, through peak nasal inspiratory flow (PNIF), quality of life (QoL), through the rhinitis quality of life questionnaire (RQLQ), the sinonasal outcome test (SNOT-22), the short-form 36-item health survey (SF-36v2), and the combined symptom medication score (CSMS), and, finally, olfactory function, through Sniffin’ sticks-16 identification test (SSIT-16), are evaluated at baseline and after treatment. Results: NARNE is the most frequent cytological pattern (48% of the total sample). The therapeutic response shows improvement in olfactory function and QoL. Conclusions: The results of this study confirm that intranasal mometasone furoate is an effective treatment for patients with NAR.

## 1. Introduction

The term “*rhinitis*” indicates a generic irritation and inflammation of the nasal mucosa characterized by rhinorrhea, blocked nose, sneezing and/or itchy nose. The following three distinct subgroups of rhinitis are commonly described: allergic rhinitis (AR), infectious rhinitis, and non-allergic, non-infectious rhinitis (NAR) [1,2]. However, it is essential to underline how frequently, in some patients, the characteristics of the above categories may overlap.

The prevalence of NAR and AR in the general population is 27% and 12%, respectively [3]. NAR is often underestimated compared to AR, although about 200 million people worldwide are affected by this condition [4].

Considering this high prevalence, chronic rhinitis has a significant social impact due to direct (drugs, medical visits, etc.) and indirect costs (absenteeism from work) [5].

Non-allergic rhinitis includes various phenotypes that may require different treatment strategies. This heterogeneous group of patients can be classified into the following several subgroups, among which are: drug-induced rhinitis, hormone-induced rhinitis, senile rhinitis, occupational rhinitis and idiopathic rhinitis [1].

For years, scientific efforts have been made to identify and distinguish the pathophysiology of these diseases, but to date, many aspects still remain unclear.

Based on the cellular inflammatory profile, NAR has also been classified as the following: non-allergic rhinitis with neutrophils (NARNE), non-allergic rhinitis with eosinophils (NARES), non-allergic rhinitis with mast cells (NARMA) and nonallergic rhinitis with eosinophils and mast cells (NARESMA) [6,7].

NARES represents the most widespread and important subgroup; data derived from different studies showed variable prevalence ranging between 10 and 35% of NAR [8,9,10]. This variability appears to be due to the lack of univocal diagnostic criteria applied in the studies. To date, the most common diagnostic method is the evaluation of the eosinophil count in a nasal smear in the absence of a demonstrable immune-mediated allergic component (negative in vivo and in vitro allergological examination) and of any other sinonasal pathology such as chronic rhinosinusitis with or without polyposis. However, the pathophysiology of NARES is unknown since the mechanisms underlying the recruitment and migration of eosinophils in the nasal mucosa still remain uncertain [11].

NARES is frequently associated with intrinsic asthma, chronic rhinosinusitis with nasal polyposis and other comorbidities such as obstructive sleep apnoea [11,12] and non-steroidal anti-inflammatory drug (NSAID)-exacerbated respiratory disease (N-ERD) [13].

Mometasone furoate is one of the most potent and effective intranasal corticosteroids available for the treatment of allergic rhinitis. In particular, this drug significantly improves symptoms, maintains high concentrations at the receptor sites within the nasal mucosa, has a rapid onset of action—normally within 4–12 h of the first dose, a low systemic absorption with a good efficacy and safety profile [14].

The aim of this study was to evaluate the impact of a second-generation intranasal corticosteroid (mometasone furoate, 200 µg b.i.d. for 15 consecutive days per month for 6 consecutive months), plus isotonic nasal saline, on the following: (1) cytologic pattern of local inflammation; (2) nasal airflow, through peak nasal inspiratory flow (PNIF) (L/min); (3) quality of life, through the Rhinitis Quality of Life Questionnaire (RQLQ), the Sino-nasal Outcome Test (SNOT-22), the Short-Form 36-Item Health Survey (SF-36v2) and the combined symptom medication score (CSMS) and (4) olfactory function, through Sniffin’ sticks-16 Identification test (SSIT-16), in a group of patients with NAR.

## 2. Materials and Methods

### 2.1. Study Design and Protocol

This was a prospective open-label observational study including non-allergic rhinitis patients at our institution Fondazione Policlinico Universitario A. Gemelli IRCCS in Rome, Italy, between January 2021 and January 2022. All patients gave their written informed consent to participate. The study protocol was approved by local Ethics Committee of our institution (Number of Protocol: ID 3744, December 2020).

We screened the following: patients aged between 18 and 65 years with clinical symptoms of non-allergic perennial rhinitis, negative skin prick test, negative specific IgE blood assays (RAST), and negative intranasal allergen provocation test for principal inhalant allergens (including house dust mites, major Italian pollens, mold, dogs/cats epithelium). Exclusion criteria were as follows: (1) treatment with other local or systematic medical treatment such as intranasal or oral corticosteroid during the previous 4 weeks, (2) positive skin prick test, (3) symptoms of chronic rhinosinusitis (CRS) with or without polyposis (nasal polyps at nasal endoscopy or evidence of sinonasal occupancy at CT scan), (4) current systemic or upper respiratory tract infections, (5) recent sinonasal surgery within the past 3 months, (6) previous diagnosis of ciliary dyskinesias, cystic fibrosis, nasal malformations or anosmia.

At baseline, patients were recruited during an outpatient allergological visit if NAR was suspected due to co-existence of clinical symptoms, negative results at allergological examination for main aeroallergens and recently performed CT scan negative for CRS. Thereafter, the patients underwent an otolaryngological visit including rhino-fibro-endoscopy and olfactometry. 

Then, all patients underwent a nasal cytologic evaluation. The following four patterns were considered: NARNE, NARES, NARESMA and NARMA.

After baseline evaluation (time zero, T0), all patients started intranasal mometasone furoate, 200 µg b.i.d. for 15 consecutive days per month plus isotonic nasal saline for 6 consecutive months.

After 6 months of treatment, the same allergological and otolaryngological tests were performed.

Patients could withdraw from the study at any time if they wished. Furthermore, patients were allowed to continue any chronic drug therapies already prescribed for other clinical reasons.

### 2.2. Procedures

#### 2.2.1. Skin Prick Tests (SPTs) and IgE

SPTs were performed using the main extracts for perennial and seasonal aeroallergens (Lofarma, Milan, Italy) on the volar part of the forearm of each patient at a distance of at least 2–3 cm from the wrist and from the antecubital fold with a distance between the different extracts ≥ 2 cm. For the puncture of the superficial layers of the skin, sterile plastic lancets with a 1 mm unilateral tip were used, which were passed through the drop of the allergenic extract and kept pressed against the skin for at least 1 s with equal pressure applied to each test. Each allergen was punctured with a new lancet to avoid the risk of contamination. A histamine solution (10 mg/mL) and a physiological saline solution (0.9%) were used as positive and negative controls, respectively. Skin prick tests were read 15–20 min after applying the extracts and the reaction was considered positive if it caused the appearance of a weal ≥ 3 mm in diameter [15].

The dosage of specific IgE (UniCAP-Phadia, Thermofisher, Uppsala, Sweden) was performed by immune-enzymatic techniques using peripheral venous sample. The following allergens were tested: dermatophagoides, aspergillus, alternaria, candida, dog and cat epithelium, staphylococcal enterotoxins, grasses, parietaria. A dosage greater than 0.35 kUA/L was considered positive. 

#### 2.2.2. Endoscopic Evaluation

Fiberoptic nasal endoscopy was performed in all subjects under local anesthesia with topical application of 2% xylocaine and using 0° and 30°, 4 mm diameter rigid nasal endoscope (Karl Storz, Tuttlingen, Germany). Nasal endoscopy was performed by using the standard three-pass technique as described by Kennedy. Nasal endoscopy findings are noted using Lund-Kennedy Endoscopic Scoring system 8 to assess the following parameters: nasal mucosa oedema (absent = 0; mild-moderate = 1 or polypoid degeneration = 2), presence of secretion (absent = 0, hyaline = 1 or thick and/or mucopurulent = 2) and presence of polyps (absent = 0, limited to the middle meatus = 1 or extended to the nasal cavity = 2), Scarring (absent = 0; mild = 1; severe = 2) and Crusting (absent = 0; mild = 1; severe = 2). The assessment was performed bilaterally, with the total points corresponding to the sum of values obtained on both sides. A total score > 4 was considered indicative of significant modification of endoscopic appearance [16].

### 2.3. Outcome Measurements

#### 2.3.1. Nasal Cytology

Nasal cytology was performed using a small curette (Nasal Scraping^®^, EP Medica, Fusignano, Ravenna, Italy) in sterile disposable plastic material, which was swiped on the patient’s nasal mucosa at the level of the middle third of the lower turbinate. The curette was rotated with small, cautious rotational and anterior-posterior movements. The collected cellular material was transferred to a microscope slide and spread over a very large area. Subsequently, the slide was dried in the air and stored in special trays. Staining of the slide was carried out by May–Grunwald–Giemsa (MGG) staining. It involved a first staining with the pure May–Grunwald dye, which was allowed to act for 3 min, a second staining with the same dye diluted 1:1 and left to act for 6 min at the end of which the slide was removed from the tray and placed in distilled water for one minute. At the end of this procedure the slides were immersed in Giemsa’s solution diluted with buffered water with a ratio of 1:10 and left in solution for about 30 min. Subsequently, the slide was removed from Giemsa’s solution, washed with a jet of running water and allowed to dry. The reading of the slides was carried out using an optical microscope equipped with an objective with a magnification capacity up to 1000×. For the analysis of the rhino-cytogram, we proceeded with a reading by fields (not less than 50) to find the cellular elements important for the diagnosis (eosinophils, mast cells, neutrophils, bacteria, spores, etc.), reporting the cell count, a semi-quantitative assessment, and the degree of inflammation according to agreed values [7].

#### 2.3.2. Peak Nasal Inspiratory Flow (PNIF)

PNIF was measured to assess the degree of patients’ nasal obstruction. For the evaluation, we used the PNIF-meter, a simple-to-use instrument with proven diagnostic validity, which measures the PNIF through the nasal cavity, providing an objective value of the degree of nasal obstruction. Values between 80 L/min and 200 L/min are considered normal, with an average physiological value of approximately 140 L/min [17].

#### 2.3.3. Olfactory Evaluation

Sniffin’ sticks-16 Identification test (SSIT-16) was performed by administering 16 odors at suprathreshold intensity to the patient. Patient identified each odor presented by choosing from the four options provided. Depending on the number of correctly identified substances, a result between 0 (no substance identified) and 16 (all substances identified) was obtained. This allowed us to classify patients as anosmic (score between 0 and 5), hyposmic (score between 6 and 10) or normosmic (score between 11 and 16) [18].

#### 2.3.4. Questionnaires 

##### Short-Form 36-Item Health Survey Italian Version 2 (SF-36v2)

The SF-36v2 is a self-reported questionnaire comprising 36 items measuring the following eight dimensions of general QoL: physical functioning (10 items), role limitation due to physical health problems (4 items), bodily pain (2 items), general health perceptions (5 items), vitality (4 items), social functioning (2 items), role limitations due to emotional problems (3 items), and general mental health (5 items). In addition to individual dimension scores, the following two summary scores assessing physical and mental dimensions of health and well-being can be calculated: physical component summary (PCS) score and the mental component summary (MCS) score, respectively. Each question’s score was coded, summed up, and transformed to a scale of 0 (worst possible health state measured by the questionnaire) to 100 (best possible health state) [19].

##### Sino-Nasal Outcome Test (SNOT-22)

SNOT-22 is composed of 22 CRS-related items scored from 0 to 5 (total score range 0–110, higher scores represent worse symptoms). SNOT-22 items consist of the following 2 categories: questions about physical symptoms (items 1–12) and questions about health and QOL (items 13–22) [20,21,22]. 

##### Rhinitis Quality of Life Questionnaire (RQLQ)

The rhinoconjunctivitis quality of life questionnaire (RQLQ) measures the problems that adults with rhinoconjunctivitis experience in their everyday lives. It has 28 questions in 7 domains (activity limitation, sleep problems, nose symptoms, eye symptoms, non-nose/eye symptoms, practical problems, and emotional function). Patients evaluated their experience during the previous week by rating each answer from 0 (no impairment) to 6 (severe impairment) [23,24]. 

##### Combined Symptom Medication Score (CSMS)

CSMS is a combination of the daily symptom score (nasal and conjunctival symptoms) and the daily medication score. It investigates the efficacy of allergic rhinitis treatment. In 2014, the EAACI Task Force for the standardization of clinical outcomes used in allergen immunotherapy trials for allergic rhinoconjunctivitis provided a homogeneous terminology for nasal and conjunctival symptoms using the six organ-related categories in the daily symptom score (dSS), stepwise use of rescue medication summed in the daily medication score (dMS) and a scoring system for a combined symptom and medication score (CSMS), which is based on an equal weight of the dSS and of the dMS. Higher values indicate worse conditions [25].

### 2.4. Statistical Analysis and Sample Size

Given the purely observational nature of the study, no a priori hypotheses were formulated; therefore, a formal calculation of the sample size was not performed. A minimum number of 30 patients to be enrolled was defined.

Continuous data were tested for normal distribution using the Kolmogorov-Smirnov test. Normally distributed and skewed variables were expressed as mean ± standard deviation (SD) and median (interquartile range), respectively. The categorical data were presented as *n* (%). The paired Student’s *t*-test and Wilcoxon signed-rank test were employed for comparisons of outcome measurements before and after treatment. Regression analysis was performed to investigate the best prediction of the dependent variables (QoL questionnaires and olfactory test) based on the data set including eosinophilic blood count, cytology, and anthropometric findings as independent variables. The β coefficient of the linear regression analyses specifies how much the dependent variable changes for a given change in each independent variable. The *p*-value < 0.05 was considered significant. Statistical analysis was performed using Stata version 9 (StataCorp LLC, College Station, TX, USA).

## 3. Results

Fifty patients fulfilling BSACI diagnostic criteria for NAR—symptoms of rhinitis without any identifiable allergic triggers—and not enrolled in other trials with oral and/or intranasal corticosteroids (INS), intranasal ipratropium, topical capsaicin, topical or oral antihistamines and montelukast, were screened. During clinical work-up, 19 were excluded as follows: 5 patients dropped out after 5 ± 1 weeks for reported subjective improvement in symptoms and 4 patients did not return at T1 for the follow-up visit. Finally, 10 patients dropped out without giving a reason. The remaining 31 subjects were included in the study. Their demographic and clinical characteristics at baseline are detailed in Table 1.

No adverse reactions or complications potentially attributable to nasal treatment were recorded.

At baseline, 15 patients (48% of the total sample) had cytologic findings of NARNE; 3 patients (10%) with the NARES pattern; 1 patient (3%) with the NARMA pattern; the NARESMA pattern was not found. Mucinous cells, evaluated as the ratio of the epithelial cells and expressed as a percentage, were significantly revealed (more than 24% of cells per 10 high power fields) in 6 patients. Finally, a normal nasal cytogram was found in the remaining 6 patients.

As shown in Table 2, the mean PNIF value was low before starting treatment (80 L/min, ±34), an expression of relevant nasal obstruction. Instead, the results of SSIT-16 revealed that almost all patients were normosmic, with only 5 hyposmic patients (mean value 12, ±2). As regards QoL, all questionnaires showed the relevant impact of NAR on patients’ well-being. In fact, the mean SF-36 value was 61 (±20, 22–91). The disease-specific QoL surveys also confirmed poor condition. The median SNOT-22 score was 45 (range 11–93, due to nonnormal distribution); the mean CSMS value was 3 (±1) and the mean RQLQ score was 2.7 (±1.1).

After treatment with the second-generation intranasal corticosteroid (mometasone furoate, 200 µg b.i.d. for 15 consecutive days per month for 6 consecutive months), plus isotonic nasal saline, the frequency distribution of cytologic phenotypes significantly changed for normal cytologic pattern (*p* = 0.0025, chi-square test), mucinous metaplasia (*p* = 0.0565, chi-square test) and NARNE (*p* = 0.0185, chi-square test). This analysis was not applicable to NARES, NARMA and NARESMA.

The intranasal corticosteroid treatment improved olfactory function (mean value of 14, ±2, *p* = 0.0005), SNOT-22 score (median value 39, 2–82, *p* = 0.0015) and RQLQ score (mean value 2.3, ±1, *p* = 0.0412). No significant differences between the two measurements of PNIF and SF-36 [Table 2].

The research of determinants of variation of outcome measurements started with the evaluation of the frequency distribution of the change (*delta*) of each outcome measurement significantly modified after treatment. Therefore, delta SSIT-16, delta SNOT-22, delta RQLQ and delta CSMS were selected as dependent variables of linear regression models.

The significant improvement of RQLQ after treatment was directly related to the he cytologic pattern of NARES at baseline (Coeff. 1.57, standard error 0.67, *p* = 0.0267) and inversely related to BMI (Coeff. –0.12, standard error 0.05, *p* = 0.0207), adjusting for blood eosinophilic count (linear regression model, *p* = 0.0281).

No other determinants were identified for the remaining changes in outcome measurements.

## 4. Discussion

In this prospective study, we found that treatment with intranasal mometasone furoate, 200 µg b.i.d., for 15 consecutive days per month for 6 consecutive months, plus isotonic nasal saline, significantly improved olfactory function and QoL in an Italian population of patients with NAR and NARNE as the most frequent cytological pattern.

To date, NAR is a growing medical problem in terms of prevalence [26,27,28,29,30], impact on patients’ quality of life [31,32,33] and health care costs [34,35].

Our study confirmed the female predominance described in the literature [36,37], but did not confirm the anamnestic relationship with smoking, a well-known irritative agent on the mucosa of the respiratory tract [38]. Consistent with the results robustly obtained by Becker et al. in 2016 [39], we found no specific IgE to *Staphylococcus aureus* enterotoxins in the sera of enrolled patients. Interestingly, our study confirmed the pioneering evidence from Gelardi et al. of a relevant sensitization to metals in NAR even in the absence of cutaneous lesions [40]. In fact, ACD was present in 8 enrolled patients and 2 patients also suffered from SNAS.

Currently, topical corticosteroid therapy is one of the pillars of the management of patients with NAR [29]. However, the small sample size, the variations in the dosages and schedule of intranasal corticosteroid administration and the heterogeneity of the study design partially explain the paucity of published meta-analyses and the low evidence supporting the role of these drugs [29].

Mometasone furoate is a second-generation intranasal corticosteroid. In 2019, Carvalho et al. compared its association with isotonic saline spray versus isotonic saline spray alone in allergic rhinitis and NAR. The study design included a total daily dose of 200 µg for two weeks. The trial was focused on improvement in nasal patency evaluated by PNIF and symptoms’ control measured with the combined nasal symptom score including nasal blockage, sneezing, nasal itching and rhinorrhea evaluated with a VAS scale from 0 to 10 (10 was worse). No superiority of association between nasal corticosteroid and isotonic saline was found [41].

Previously, Lundblad et al. conducted a phase III, double-blind, randomized, placebo-controlled, Nordic multicenter study enrolling 329 patients with NAR treated with mometasone furoate (total daily dose of 200 µg) for six weeks. The authors showed that the drug was safe and effective for patients with NAR [42].

In our pilot study, we used the same dosage as in previously mentioned trials, extending the period of treatment to 6 months as suggested in a recent meta-analysis [29]. Furthermore, we have provided a background therapy, represented by saline solution, in order to guarantee constant irrigation and nasal hygiene during the entire study period and not only during the phases of taking mometasone furoate. Finally, we expanded the outcomes by incorporating both objective measures of nasal respiratory (PNIF) and olfactory (SSIT-16) functions and self-reported outcomes regarding the impact of therapy on QoL (SF-36, SNOT-22, RQLQ and CSMS).

The initial endotypization of patients according to nasal cytology revealed a predominance of NARNE.

To the best of our knowledge, this is the first real-life study showing positive effects of mometasone furoate, 200 µg b.i.d. for 15 consecutive days per month for 6 consecutive months, plus isotonic nasal saline, on olfactory function (SSIT-16) and health-related QoL evaluated through disease-specific questionnaires (SNOT-22, RQLQ and CSMS). Our results confirm previous evidence that QoL is significantly impaired in patients with NAR, especially in the case of the NARES pattern [31].

Notably, the evaluation of the determinants of change of outcome measurements significantly modified after local steroid treatment revealed a predictive role of cytologic NARES pattern at baseline for the improvement of RQLQ. Furthermore, an inverse relationship with BMI was found, independently of the blood eosinophilic count. We can speculate that the predictive role of the NARES endotype could be partially explained by multiple similarities with well-characterized type 2 diseases, such as eosinophilic asthma and chronic rhinosinusitis with nasal polyps (CRSwNP), conditions that are generally well corticosteroid-responsive [11].

This study has some weaknesses. The following major criticism could be made of the sample size and the drop-out rate: 19 of 51 patients (37%) dropped out of the study. The main reason given by the patients who dropped out of the study was the subjective improvement in symptoms after a few weeks of treatment. Secondly, due to the purely observational nature of the trial, we did not compare the treatment with mometasone furoate plus isotonic nasal saline versus placebo/other control arms. We preferred to perform a broad multidisciplinary evaluation for each enrolled patient, identifying the specific endotype, using appropriate diagnostic methods (SPTs, blood testing for specific IgE in serum) and proper ear–nose–throat (ENT) assessment. Furthermore, we focused our attention on outcomes important to patients with NAR, adopted validated tools to measure them and defined a relatively long study period, as suggested in a recent review by the Cochrane ENT group [29].

## 5. Conclusions

This study enriches the literature on the efficacy of intranasal corticosteroid therapy with mometasone furoate in patients with nonallergic rhinitis. 

## Figures and Tables

**Table 1 jpm-12-01179-t001:** The baseline socio-demographics and clinical characteristics of the participants.

	Patients (*n* = 31)	
Variables	*N*	%
Female gender	24	77
Age [mean ± SD ^~^ (range)]	45 (±13, 18–64)	
BMI * [kg/m^2^, mean ± SD ^~^ (range)]	24 (±4, 17–35)	
Social class		
Employed	23	74.2
Unemployed	2	6.5
Housewife	2	6.5
Student	4	12.8
Active current smokers	4	12.9
Ex-smokers	6	19.4
Nonsmokers	21	67.7
Family history of allergy	13	42
Non-respiratory allergies	12	39
Food allergy	1	3 ’
NSAID ^†^ hypersensitivity	2	7 ’
Hymenoptera allergy	1	3 ’
ACD ^∫^	8	26 ’
SNAS ^”^	2	7 ’
Asthma	3	10
Total serum IgEs [kU/L, mean ± SD ^~^ (range)]	63 (±124, 3–691)	
Previous nasal surgery	6	19
Eosinophils [n/µL, mean ± SD ^~^ (range)]	155 (±112, 30–500)	
Specific IgE to *Staphylococcus aureus* enterotoxins [kU/L, mean ± SD ^~^ (range)]	0	

* BMI, body mass index. ^~^ SD, Standard Deviation. ’, % of total sample. ^†^ NSAID, non-steroidal anti-inflammatory drugs. ^∫^ ACD, allergic contact dermatitis.” SNAS, systemic nickel allergy syndrome.

**Table 2 jpm-12-01179-t002:** Change of outcome measurements after treatment.

Outcome Measurement	At Baseline (T0)	After Treatment (T1)	*p*-Value of the Change
Peak Nasal Inspiratory Flow (PNIF) [L/min., mean value, ±SD *]	80 [±34]	89 [±38]	N.S. ’
Sniffin’ sticks-16 Identification test (SSIT-16) [mean value, ±SD *]	12 [±2]	14 [±2]	0.0005 ’
SF-36 [mean value, ±SD *]	61 [±20]	65 [±20]	N.S. ’
SNOT-22 [median value, range]	45 [11–93]	39 [2–82]	0.0015 ^†^
RQLQ [mean value, ±SD *]	2.7 [±1.1]	2.3 [±1]	0.0412 ’
CSMS [mean value, ±SD *]	2.7 [±1.3]	3.3 [±1.1]	0.0543 ’

* SD, standard deviation. ’, evaluation of the change between pre- and post-treatment through paired Student’s *t*-test due to normal distribution of the variables. ^†^, evaluation of the change between pre- and post-treatment through Wilcoxon signed-rank test due to non-normal distribution of the variable.

## Data Availability

Not applicable.

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
