# Peer review of "Mometasone Furoate in Non-Allergic Rhinitis: A Real-Life Italian Study"

_jpm, 2022, doi:10.3390/jpm12071179_

Round 1
Reviewer 1 Report
clinically important informations on mometazo
ne froate efficacy on non allergic rhinitis are written in this manuscript. I think contents of this multicentral clinical study is worthwhile for clinicians.
Author Response
July, 12th 2022
To Editor and Reviewers
Journal of Personalized Medicine MDPI
We would like to greatly thank the Editor and Reviewers who encouraged a revision of the manuscript.
Please find the enclosed the Revision vers. 1 of the Original article entitled “Mometasone Furoate in Non-Allergic Rhinitis: a Real-Life Italian Study” by Angela Rizzi, Giuseppe Parrinello, Eugenio De Corso, Laura Tricarico, Michele Centrone, Alessia Di Rienzo, Chiara Laface, Giulio Cesare Passali, Gabriella Cadoni, Riccardo Inchingolo, Gaetano Paludetti, Jacopo Galli and Eleonora Nucera.
[Journal of Personalized Medicine] Manuscript ID: jpm-1820658- Minor Revisions
Author's Reply to the Review Report (Reviewer 1)
Comments and Suggestions for Authors
Clinically important information on mometasone furoate efficacy on non allergic rhinitis are written in this manuscript. I think contents of this multicentral clinical study is worthwhile for clinicians.
We thank the Reviewer for the appreciation.
With the best regards,
Angela Rizzi, Giuseppe Parrinello, Eugenio De Corso, Laura Tricarico, Michele Centrone, Alessia Di Rienzo, Chiara Laface, Giulio Cesare Passali, Gabriella Cadoni, Riccardo Inchingolo, Gaetano Paludetti, Jacopo Galli and Eleonora Nucera
Corresponding Author:
Riccardo Inchingolo, MD, PhD
UOC Pneumologia, Fondazione Policlinico Universitario A. Gemelli IRCCS. Largo A. Gemelli, 8 – 00168 – Rome, Italy.
riccardo.inchingolo@policlinicogemelli.it
Corresponding Author will receive all editorial communications
The authors declare that the manuscript, or specified parts of it, have not been and will not be submitted elsewhere for publication.
Reviewer 2 Report
This study is very interesting trial using mometasone furoate in NAR. But, I wonder why you used mometasone furoate plus isotonic saline irrigation instead of using mometasone furoate only. According to the reference of Carvalho (#41), mometasone furoate is not superior to saline. The reason of using mometasone furoate plus isotonic saline irrigation should be more clarified.
Author Response
July, 12th 2022
To Editor and Reviewers
Journal of Personalized Medicine MDPI
We would like to greatly thank the Editor and Reviewers who encouraged a revision of the manuscript.
Please find the enclosed the Revision vers. 1 of the Original article entitled “Mometasone Furoate in Non-Allergic Rhinitis: a Real-Life Italian Study” by Angela Rizzi, Giuseppe Parrinello, Eugenio De Corso, Laura Tricarico, Michele Centrone, Alessia Di Rienzo, Chiara Laface, Giulio Cesare Passali, Gabriella Cadoni, Riccardo Inchingolo, Gaetano Paludetti, Jacopo Galli and Eleonora Nucera.
[Journal of Personalized Medicine] Manuscript ID: jpm-1820658- Minor Revisions
Author's Reply to the Review Report (Reviewer 2)
Comments and Suggestions for Authors
This study is very interesting trial using mometasone furoate in NAR. But, I wonder why you used mometasone furoate plus isotonic saline irrigation instead of using mometasone furoate only. According to the reference of Carvalho (#41), mometasone furoate is not superior to saline. The reason of using mometasone furoate plus isotonic saline irrigation should be more clarified.
We thank the Reviewer for the comment. We have provided a background therapy, represented by nasal saline solution, in order to guarantee constant irrigation and nasal hygiene during the entire study period and not only during the phases of taking mometasone furoate. We added this sentence in “Discussion” section. Lines: 316-319.
With the best regards,
Angela Rizzi, Giuseppe Parrinello, Eugenio De Corso, Laura Tricarico, Michele Centrone, Alessia Di Rienzo, Chiara Laface, Giulio Cesare Passali, Gabriella Cadoni, Riccardo Inchingolo, Gaetano Paludetti, Jacopo Galli and Eleonora Nucera
Corresponding Author:
Riccardo Inchingolo, MD, PhD
UOC Pneumologia, Fondazione Policlinico Universitario A. Gemelli IRCCS. Largo A. Gemelli, 8 – 00168 – Rome, Italy.
riccardo.inchingolo@policlinicogemelli.it
Corresponding Author will receive all editorial communications
The authors declare that the manuscript, or specified parts of it, have not been and will not be submitted elsewhere for publication.